# Portable Rabies Virus Sequencing in Canine Rabies Endemic Countries Using the Oxford Nanopore MinION

**DOI:** 10.3390/v12111255

**Published:** 2020-11-04

**Authors:** Crystal M. Gigante, Gowri Yale, Rene Edgar Condori, Niceta Cunha Costa, Nguyen Van Long, Phan Quang Minh, Vo Dinh Chuong, Nguyen Dang Tho, Nguyen Tat Thanh, Nguyen Xuan Thin, Nguyen Thi Hong Hanh, Gati Wambura, Frederick Ade, Oscar Mito, Veronicah Chuchu, Mathew Muturi, Athman Mwatondo, Katie Hampson, Samuel M. Thumbi, Byron G. Thomae, Victor Hugo de Paz, Sergio Meneses, Peninah Munyua, David Moran, Loren Cadena, Andrew Gibson, Ryan M. Wallace, Emily G. Pieracci, Yu Li

**Affiliations:** 1Poxvirus and Rabies Branch, Division of High-Consequence Pathogens and Pathology, National Center for Emerging and Zoonotic Infectious Diseases, Centers for Disease Control and Prevention, Atlanta, GA 30329, USA; lzu1@cdc.gov (C.M.G.); hws5@cdc.gov (R.E.C.); euk5@cdc.gov (R.M.W.); ydi7@cdc.gov (E.G.P.); 2Mission Rabies, Tonca, Panjim, Goa 403001, India; gowri@missionrabies.com; 3Disease Investigation Unit, Directorate of Animal Health and Veterinary Services, Patto, Panjim, Goa 403001, India; niceta.cunhacosta@gmail.com; 4Vietnam Department of Animal Health, Hanoi 100000, Vietnam; long.dahvn@gmail.com (N.V.L.); phanquangminh1@gmail.com (P.Q.M.); vodinhchuong3@gmail.com (V.D.C.); 5National Center for Veterinary Diseases, Hanoi 100000, Vietnam; thovet99@yahoo.com; 6Sub-Department of Animal Health, Phú Thọ Province 35000, Vietnam; nguyentatthanh63@gmail.com (N.T.T.); nxtsubdah@gmail.com (N.X.T.); nguyenhonghanh.typt@gmail.com (N.T.H.H.); 7Center for Global Health Research, Kenya Medical Research Institute, Nairobi 00100, Kenya; gati.wambura@wsu.edu (G.W.); FAde@kemricdc.org (F.A.); OMito@kemricdc.org (O.M.); chuchumbaire@gmail.com (V.C.); thumbi.mwangi@wsu.edu (S.M.T.); 8Department of Public Health, Pharmacology and Toxicology, University of Nairobi, Nairobi 00100, Kenya; 9Zoonotic Disease Unit, Ministry of Health, Ministry of Agriculture, Livestock and Fisheries, Nairobi 00100, Kenya; muturimathew@gmail.com (M.M.); amwatondo@yahoo.com (A.M.); 10Institute of Biodiversity, Animal Health and Comparative Medicine, University of Glasgow, Glasgow G12 8QQ, UK; Katie.Hampson@glasgow.ac.uk; 11University of Nairobi Institute of Tropical and Infectious Diseases, Nairobi 00100, Kenya; 12Paul G. Allen School for Global Animal Health, Washington State University, Pullman, WA 99164, USA; 13Ministry of Agriculture Livestock and Food, Guatemala City 01013, Guatemala; byronthomae@yahoo.es; 14National Health Laboratory, MSPAS, Villa Nueva 01064, Guatemala; depaz.victor@lns.gob.gt (V.H.d.P.); sergiomeneses1@yahoo.com (S.M.); 15Division of Global Health Protection, Centers for Disease Control, Nairobi 00100, Kenya; ikg2@cdc.gov; 16University del Valle de Guatemala, Guatemala City 01015, Guatemala; dmoran@CES.UVG.EDU.GT; 17Division of Global Health Protection, Centers for Disease Control, Guatemala City 01001, Guatemala; bqp3@cdc.gov; 18The Roslin Institute and The Royal (Dick) School of Veterinary Studies, Division of Genetics and Genomics, The University of Edinburgh, Easter Bush Veterinary Centre, Roslin, Midlothian EH25 9RG, UK; andy@missionrabies.com

**Keywords:** rabies, lyssavirus, canine rabies elimination, portable sequencing, nanopore, MinION

## Abstract

As countries with endemic canine rabies progress towards elimination by 2030, it will become necessary to employ techniques to help plan, monitor, and confirm canine rabies elimination. Sequencing can provide critical information to inform control and vaccination strategies by identifying genetically distinct virus variants that may have different host reservoir species or geographic distributions. However, many rabies testing laboratories lack the resources or expertise for sequencing, especially in remote or rural areas where human rabies deaths are highest. We developed a low-cost, high throughput rabies virus sequencing method using the Oxford Nanopore MinION portable sequencer. A total of 259 sequences were generated from diverse rabies virus isolates in public health laboratories lacking rabies virus sequencing capacity in Guatemala, India, Kenya, and Vietnam. Phylogenetic analysis provided valuable insight into rabies virus diversity and distribution in these countries and identified a new rabies virus lineage in Kenya, the first published canine rabies virus sequence from Guatemala, evidence of rabies spread across an international border in Vietnam, and importation of a rabid dog into a state working to become rabies-free in India. Taken together, our evaluation highlights the MinION’s potential for low-cost, high volume sequencing of pathogens in locations with limited resources.

## 1. Introduction

Rabies is a progressive, fatal encephalitis caused by members of 17 different viral species within the *Lyssavirus* genus of rhabdoviruses [1]. The vast majority of human rabies deaths are caused by a virus in the species *Rabies lyssavirus* [2], commonly referred to as rabies virus. Current estimates suggest over 59,000 people die of rabies each year across the world [3]. More than 99% of human rabies cases are caused by rabid dogs, and control of rabies in dogs can reduce human cases [2,3]. Canine rabies has been effectively controlled and even eliminated in several countries through coordinated surveillance and canine vaccination campaigns [2,4,5,6,7,8,9]. The World Health Organization (WHO), the World Organization for Animal Health (OIE) and the Food and Agriculture Organization (FAO) have set a goal to eliminate dog-mediated human rabies deaths by 2030 [10]. However, many countries with endemic canine rabies are in the early stages of planning control efforts and face barriers including limited understanding of rabies prevalence, logistical challenges, and competition for limited resources [9,11].

The number of rabid dogs and human rabies deaths is likely underreported in many canine rabies endemic countries [2,12,13,14,15,16,17]. The absence of reliable surveillance data makes it challenging to estimate the rabies burden, procure funding and plan control strategies [3,17], and lack of data has been a major factor in the low prioritization of rabies in endemic countries [17]. Rabies surveillance using validated laboratory tests is a critical component of rabies control and elimination efforts [2,11,18,19,20,21,22]. Good surveillance data are required to gain an accurate estimate of the rabies prevalence so resources can be allocated in the most effective way possible [19,23,24]. As rabies elimination efforts progress, continued surveillance is necessary to determine the impact of control efforts, confirm canine rabies elimination, and promptly identify re-emergence [11,18,19,21,24,25].

It is important to not only identify the presence of rabies in a region but also to characterize the circulating variants, determine which host species are responsible for virus maintenance, and identify the geographic distribution of rabies across different regions and borders. Rabies virus can infect many mammalian species but has established sustained transmission cycles in only a few host species [26,27,28,29,30]. Rabies virus has adapted to mammalian reservoir species including domestic dogs and wildlife over hundreds of years of evolution [31,32,33], and rabies virus circulating in different reservoirs or regions can be distinguished genetically or antigenically [31,34,35,36,37,38,39]. Different variants of rabies virus and even different species of lyssavirus can overlap in distribution, co-exist in the same country, or span borders [40,41,42,43,44,45,46]. Establishing laboratory techniques to quickly identify different rabies virus variants is important to help plan and optimize the effectiveness of control efforts and responses to cases [25]. For instance, control strategies differ greatly for rabies circulating in domesticated dogs compared to vampire bats. Rabies virus gene sequencing is one method that can provide high resolution information to distinguish rabies virus variants. Moreover, sequencing data can be combined with other surveillance data to determine the host reservoir and geographic limits of viral circulation.

Rabies virus has a single-stranded, non-segmented negative-sense RNA genome that is approximately 12 kilobases and encodes five proteins: a nucleoprotein, phosphoprotein, matrix protein, glycoprotein, and an RNA-dependent RNA polymerase [47]. The nucleoprotein is highly conserved and is most abundant during rabies virus infection [34,47]. The nucleoprotein gene is routinely sequenced around the world for rabies virus genotyping or phylogenetic analysis, and most publicly available rabies virus sequences are partial or complete nucleoprotein gene sequences. The glycoprotein is involved in receptor binding, viral entry, immune response, and host adaptation [48,49,50]. Glycoprotein gene sequencing is routinely performed during studies of rabies virus phylogeny, host adaptation, immunogenicity, and viral evolution. Combined sequencing of both the nucleoprotein and glycoprotein genes allows for comparison with many reference sequences as well as increased resolution compared to sequencing either gene on its own.

Sequencing can provide valuable insight into rabies in an area; however, until recently, access to sequencing has been limited to high-income settings. First- or second-generation sequencing technologies require large machines with expensive service contracts and expertise in sample preparation and data analysis. The cost of sequencing has dramatically decreased, and much more sequence can now be produced for a much lower cost [51,52,53,54]. In developed countries, sequencing can be easily performed either in house at a scientific institute or by shipping samples to a third party. However, the initial investment in high-cost platforms and inaccessibility to proper training has prevented sequencing in developing countries. Improvements in the cost, ease of use, portability, and availability of reagents for rabies diagnostic testing and characterization will make it easier to initiate effective rabies control programs in resource-limited settings [21,55]. The potential for sequencing in resource-limited areas is high because molecular techniques are well adapted for testing in remote locations. Commercially available products offer long-term storage of nucleic acids at room temperature [56,57,58], so samples can be collected in the field, stabilized, and transported or shipped to a laboratory without the need for cold chain. Technical improvements have led to a new generation of rapid, lower cost, portable point-of-care testing devices, including PCR machines and sequencers [59,60,61]. The Oxford Nanopore MinION is a portable sequencer that is much smaller and less expensive than traditional sequencers, allowing for sequencing outside of traditional molecular laboratories [62,63].

We developed and validated a portable strategy to sequence rabies virus nucleoprotein and glycoprotein genes in rabies diagnostic laboratories in canine rabies endemic countries using the Oxford Nanopore MinION sequencer. Rabies virus genes were sequenced in public health laboratories in Guatemala, India, Kenya, and Vietnam. In total, 259 sequences were produced. Phylogenetic analysis and comparative sequence analysis were performed to determine rabies virus variant, importation/translocation events, examine geographical distribution, and identify clustering. Sequences and interpretations were shared with the local rabies surveillance systems in each respective country and were made publicly available.

## 2. Materials and Methods

### 2.1. Samples

All samples were collected from postmortem brains as part of routine rabies surveillance activities. Human samples were de-identified prior to testing. No animal or human sampling was performed for this study. All diagnoses and clinical outcomes were determined per the routine operation of the participating laboratory and were not based on the results of this study. Rabies cases were identified through antibody staining of brain tissue following the local guidelines for the direct fluorescent antibody test (DFA/FAT) [64] or LN34 real-time RT-PCR [65,66]. Detailed sample information can be found in Appendix A.

### 2.2. Nanopore Sequencing

Total RNA was extracted from rabies positive brain samples using the Direct-zol RNA miniprep kit (R2051 Zymo, Irvine, CA, USA) as previously described [65]. cDNA was generated in a 25 µL reaction using 10 µL RNA, 1.5 µL 100 µM random hexamer oligonucleotides, 0.5 µL Protector RNase Inhibitor (3335399001 Roche, Sigma-Aldrich, St. Louis, MO, USA), and Roche AMV reverse transcriptase (10109118001 Roche). Reverse transcription was run at 42 °C for 90 min followed by heat inactivation for 10 min at 95 °C. cDNA was then diluted by adding 75 µL nuclease-free water to each sample. A volume of 5 µL diluted cDNA was added to the primary PCR reaction mix containing 1 µL each of 20 µM forward and reverse primers; PCR was performed using Takara long amplicon Taq with GC buffers (RR02AG Takara Bio USA, Mountain View, CA, USA). Primer sequences were designed with barcoding tags according to the Oxford Nanopore protocol for PCR barcoding of 96 samples (EXP-PBC096 Oxford Nanopore Technologies, Oxford, UK); primer sequences are listed in Table 1. Nucleoprotein gene primary PCR reaction conditions were: 95 °C for 2 min; 45 cycles of 95 °C for 30 s, 58 °C for 40 s, 72 °C for 2 min 30 s, then 10 min at 72 °C. Glycoprotein gene PCR conditions were the same, except annealing was performed at 50 °C. Nucleoprotein and glycoprotein PCR products for the same sample were then combined into one tube and underwent clean-up using 0.65x AmpureXP beads (Beckman Coulter, Indianapolis, IN, USA). After washing twice in 70% ethanol, DNA was eluted off the beads with 30 µL nuclease-free water. A volume of 2 µL of cleaned PCR product was then used in the barcoding PCR reaction following the manufacturer’s instructions (EXP-PBC096 Oxford Nanopore Technologies) but using 20 µL reaction volumes and Takara long amplicon Taq with GC buffers (RR02AG Takara Bio USA). PCR conditions were: 95 °C for 2 min, 45 cycles of 95 °C for 30 s, 62 °C for 30 s, 72 °C for 2 min 30 s, then 10 min at 72 °C. During PCR barcoding, each sample receives a unique barcode that can be used to separate reads by sample bioinformatically. After barcoding, samples were quantified using Qubit dsDNA broad range assay (Q32850 Invitrogen, ThermoFisher Scientific, Waltham, USA), and samples were pooled to normalize DNA concentration per sample. Library preparation was performed using approximately 1 µg pooled DNA after clean-up using 0.65x AmpureXP beads, following instructions for the Ligation Sequencing Kit 1D (SQK-LSK108 Oxford Nanopore Technologies, Oxford, UK). Sequencing was performed overnight on a MinION sequencer using a MIN107 R9 SpotON flow cell that had >1000 pores.

### 2.3. Sequence Data Analysis

Basecalling and demultiplexing were performed using Albacore version 2.3.0 read_fast5_basecaller.py specifying barcoding, FLO-MIN107 flow cell, kit SQK-LSK108, and both fast5 and fastq output formats. All fastq files were combined into a single file for each barcode. In total, 65 nucleotides (nt) were trimmed from the end of each read using seqtk [67] to remove adapter and primer sequences. Sequences were aligned to reference rabies virus genomes using bwa mem -x ont2d. Consensus sequences were generated from sam mapping files in CLC genomics workbench version 11 (Qiagen, Venlo, Netherlands) using a minimum threshold of 50× minimum read depth, inserting “N” at any position where read depth was <50x, and vote to resolve conflicts. Consensus sequences were corrected using Nanopolish version 0.10.2 (https://github.com/jts/nanopolish/). Briefly, reads were indexed using the nanopolish index, then reads were mapped to the consensus sequence using bwa mem -x ont2d. Next, polishing was performed using the following command:

python ~/nanopore/bin/nanopolish-master/scripts/nanopolish_makerange.py consensus.fasta | parallel --results nanopolish.results -P 8 \nanopolish variants --consensus -o polished.{1}.vcf -w {1} -r reads.fastq.gz -b mapped.sorted.bam -g consensus.fasta -t 4 --min-candidate-frequency 0.1.

A polished consensus was generated using the command:

nanopolish vcf2fasta -g consensus.fasta polished.*.vcf > consensus_polished.fa.

### 2.4. Manual Homopolymer Indel Correction

Consensus and polished consensus sequences were aligned to an annotated rabies virus reference sequence (specified in Appendix A) using mafft v7.308 [68,69] in geneious 9.1.4 (Biomatters, Inc., Newark, NJ, USA). Nucleoprotein and glycoprotein coding regions were extracted. Alignments were manually inspected for indels relative to the reference sequence. Reference sequences were used to identify the location of indels but were not used to determine which base to insert during error correction. Indels in homopolymers were corrected using the following guidelines. If an indel occurred in only the consensus or the polished corrected sequence, the sequence that had no indel relative to the reference was used. For insertions in a homopolymer, one of the homopolymer bases was removed. For a deletion in a homopolymer, the base to insert was determined by the homopolymer present in the consensus sequence. In cases where indels were detected between two neighboring homopolymers, an ambiguous base was used. In all cases, correction was performed based on the homopolymer sequence of the consensus not the sequence of a reference. When a non-homopolymer nucleotide difference was observed between the consensus and the polished consensus, the nucleotide in the polished consensus was chosen. Examples of indel correction can be found in Appendix A. All sequences generated as part of this study were submitted to GenBank (accession numbers: MW054941-MW055234 and MW074784-MW074787).

### 2.5. Analysis of Sequence Accuracy

A subset of either 100, 200, 500, or 1000 reads was extracted from trimmed fastq files using seqtk [67] to generate mapping files with a range of read depth. The subset of reads was then mapped to rabies virus reference genomes using bwa mem -x ont2d. The resultant sam mapping files were used to generate consensus sequences as described above. Minimum read depth was defined as the minimum number of reads at any position in the nucleoprotein or glycoprotein gene coding region based on mapping files. Sequences were polished using Nanopolish and manual error correction of indels was performed as described above.

### 2.6. Sanger Sequencing

Nucleoprotein gene sequencing reaction was performed using the standard Sanger method using overlapping primers (Lys 001-1066degB and 550F-304), as previously described [70]. Forward and reverse primers for each set were used with a BigDye Terminator v1.1 Cycle sequencing kit. Amplicons were purified using ExoSAP-IT (USB Products Affymetrix, Inc., Cleveland, OH, USA) and sequenced on a 3730 DNA Analyzer (Applied Biosystems, Gran Island, HY, USA). Sequences were edited using Bioedit 7.0.5.3 [71] and assembled based on the N gene sequence of the SADB19 rabies virus strain (GenBank accession M31046).

### 2.7. Consensus Accuracy

Consensus, polished consensus, and manually corrected consensus nucleoprotein sequences were aligned to nucleoprotein sequences generated using Sanger sequencing. Alignments were generated using mafft v7.308 [68,69] in geneious 9.1.4 (Biomatters, Inc). Percent identity and error were determined in geneious. Determination of error type (insertion, deletion, or substitution) was performed by visual inspection of alignments.

### 2.8. Portable Sequencing

Completely portable sequencing was performed in Goa, India, in the Disease Investigation Unit (DIU), Directorate of Animal Health and Veterinary Services. The veterinary diagnostic laboratory did not have capacity for sequencing or PCR during July 2018. All equipment and reagents were transported. Portable equipment used is listed in Table 2. For Kenya, Vietnam, and Guatemala, laboratories were equipped for PCR, so portable PCR equipment was not used. Portable sequencing was performed at all locations, and preliminary analysis was performed on a Dell Precision 7720 laptop.

### 2.9. Phylogenetic Analysis

All sequence alignments were generated using mafft v7.308 in geneious. Sequence alignments were trimmed to include only the coding region of the nucleoprotein and glycoprotein genes. Phylogenetic analysis by Maximum Likelihood was performed in Mega 7.0.26 [72] using the GTR+G+I model. In total, 1000 bootstrap replicates were performed for each analysis. Trees were rooted using sequences from the following references: JQ685963 and JQ685945 (Kenya), JQ685945 and JQ685901 (India), GU992314 (Guatemala), and KF155006 (Vietnam). References from a collapsed vampire bat clade in Guatemala tree included KM594042, KM594040, KM594041, AF070449, KF864397, KC758863, KF864322, KC758862, KC758861, AB519642, GU592648, KM594041, KX148100, and KU523255.

### 2.10. Data Visualization

Data visualization and manipulation were performed in RStudio with R version 3.6.1 [73,74]. Shape files for maps were downloaded from gadm.org using the raster package [75]. Data points on maps were made at the approximate center of the location provided; however, in some cases, the resolution of location information was limited to state. Plots and maps were made using ggplot2 [76]. Figures were finished in Inkscape (inkscape.org).

### 2.11. Cost Estimate

All price estimates were based on prices advertised on the manufacturer’s websites in US dollars for ordering from within the United States. Flow cell cost of USD 500.00 per flow cell is based on bulk ordering price for 48 flow cells. Shipping costs for reagents and transportation costs for luggage were not included in this analysis. Cost to implement nanopore sequencing was not considered as cost would vary greatly depending on the existing resources of the laboratory, including access to PCR machines, computational resources for sequencing and data analysis, electronic storage of sequence data, and personnel with bioinformatics and wet lab training. Cost of tubes, pipette tips, personal protective equipment, water, and ethanol were considered negligible and were not included. Cost of basic molecular biology lab equipment (centrifuge, PCR machine, pipettes) was not considered in this estimate; cost of specialized equipment (laptop, magnetic bead stand) was also not included. Prices may not reflect prices at the time of publication and may vary substantially for different locations.

## 3. Results

### 3.1. Study Design

Rabies is a neglected zoonotic disease, and cost is a major limiting factor when implementing new approaches in control and surveillance efforts. We developed a rabies virus sequencing strategy focused on low cost and high throughput. The Oxford Nanopore MinION was chosen as the sequencing platform due to its portability and low cost. A partial genome amplicon-based approach using two-step RT-PCR was employed for amplifying rabies virus nucleic acid above the host signal. After cDNA synthesis using random hexamer primers, the complete nucleoprotein (N) and glycoprotein (G) genes were amplified by PCR using degenerate rabies-specific primers (Table 1). Amplicons for each sample were then pooled and given a unique barcode identifier by PCR, allowing up to 96 samples to be combined in a single sequencing run.

Based on this strategy, a cost estimate for nanopore sequencing of rabies virus isolates using the MinION was performed. Cost to implement nanopore sequencing was not considered as cost would vary greatly depending on the existing resources of the laboratory, including (but not limited to) access to basic molecular biology lab equipment, computational resources, and personnel with bioinformatics and molecular biology wet lab experience. The cost of RNA extraction, reverse transcription, and primary PCR was USD 6.53 per sample using the manual column extraction and reagents listed in the Methods (Appendix A). Total RNA extraction, reverse transcription, and long amplicon PCR are standard methods, and this portion of the protocol should be adaptable to changes in methodology or reagents, although a high-fidelity Taq polymerase is recommended. The cost of multiplexing, library preparation, and sequencing on the Oxford Nanopore MinION was also estimated. As with most next generation sequencing approaches, the cost decreased dramatically as more samples were multiplexed into a single sequencing run. When 50 samples were multiplexed on a single flow cell, the cost per sample was USD 14.90 (Table 3). The cost for 20 samples was USD 33.68 per sample, and the cost to sequence a single sample was USD 626.13 (Table 3, Appendix A).

### 3.2. Validation

The MinION has key advantages over other sequencing technologies in price and portability; however, sequence error rates are much higher, which is a major concern. Single raw reads produced from MinION sequencing have error rates as high as 15% using flow cells with R9.4 or R9.5 pores [77,78,79,80,81]. The majority of these sequencing errors are random and are corrected by high read depth and coverage, resulting in final consensus sequences with <1% error [80,82,83]. However, even with high read depth, consensus accuracy from MinION-derived sequences is lower than other sequencing methods such as Sanger and Illumina due to issues resolving homopolymers (runs of the same base). Even relatively short homopolymers (three to five of the same base in a row) present issues for MinION sequencing [80,81,82]. We compared the accuracy of sequencing rabies virus using the MinION to the gold standard in the rabies field, Sanger sequencing.

A total of 37 rabies positive brain samples were chosen for complete nucleoprotein gene sequencing by both Sanger and MinION sequencing. Sample details can be found in Appendix A. MinION sequencing produced 100% coverage of the complete nucleoprotein gene with over 100x read depth for all samples (Appendix A). Sanger sequencing produced complete nucleoprotein gene sequences for 31 samples and partial sequences for 4 samples. We used this dataset to examine the effect of read depth on MinION sequence consensus accuracy. To examine the effect of low read depth on consensus sequence accuracy, subsets of either 100, 200, 500, or 1000 reads were taken from the original sequencing data, resulting in 70 unique datasets with minimum read depth ranging from 4 to 59,000x. A consensus sequence was generated for each of the 70 sample datasets.

We then employed methods of error correction to reduce the high error rate of MinION consensus sequences and generate the highest accuracy sequence. The 70 consensus sequences were each error corrected using Nanopolish (available from: https://github.com/jts/nanopolish) to produce a polished consensus. After Nanopolish correction, the polished consensus sequences still contained single nucleotide insertions and deletions (indels) in homopolymer regions. Indels in homopolymer regions were manually corrected to generate a corrected consensus. Although manual correction of sequencing errors is usually avoided, we justified its use in this case for the following reasons. The rabies virus nucleoprotein gene is essential, highly conserved and is exactly 1353 nucleotides long in all known rabies virus variants. A single nucleotide indel in a homopolymer would cause a frame shift and disrupt the coding region. The likelihood of a frame shift mutation naturally occurring in the nucleoprotein gene in a viable virus is exceedingly low. Given these observations, we reasoned that manual correction of indels in homopoymers was appropriate within the coding region of the nucleoprotein gene in rabies virus. To manually correct indels in homopolymer regions, the nucleoprotein coding region of the raw and polished consensus sequences were aligned to a publicly available rabies virus reference nucleoprotein gene sequence. Indels in the polished consensus sequence were identified by comparison with the reference. These indels were corrected by either removing or adding an additional base in the homopolymer. Reference sequences were not used to determine which base to add to correct indels; single nucleotide differences between the sequence and reference were not corrected (additional details can be found in the Methods section). Sequence accuracy of the consensus, polished consensus, and manually corrected consensus sequences were determined by comparison with Sanger sequences.

The average error rate for all 70 datasets including all read depth and coverage levels was 0.16% for the raw consensus sequences, 0.07% for polished consensus sequences, and 0.01% for manually corrected sequences (Table 4). Not only did the accuracy differ, but the type of errors also differed. Raw consensus sequences contained mostly deletions and included incorrectly called bases (substitutions) relative to Sanger sequences (Figure 1). Polishing the consensus decreased the number of deletions and miscalled bases but increased the number of insertions. The manually corrected consensus sequences did not contain any insertions or deletions; remaining errors were miscalled bases (Figure 1).

The consensus accuracy increased with read depth for raw, polished, and manually corrected consensuses (Table 4, Figure 1 and Appendix A). For raw consensus and polished consensus sequences, the accuracy approached 99.96% when minimum read depth was >50x, while accuracy for raw consensus sequences was only 99.21% (99.78% for polished) when minimum read depth was <50x Notably, all incorrectly called bases in raw consensus sequences were corrected by Nanopolish when minimum read depth was >50x (Figure 1). At levels higher than 50x minimum read depth, the consensus accuracy was not greatly improved by increased read depth (Table 4). This is likely because systematic errors in homopolymer regions cannot be resolved by increased read depth. Manual correction of homopolymer indels eliminated these errors, achieving 99.996% accuracy for all samples with >50x minimum read depth (Table 4). No manually corrected consensus sequence generated from data with >50x read depth had any insertions, deletions, or miscalled bases (Figure 1). The remaining 0.004% error was due to indels between two homopolymers. In such cases, an ambiguous base representing both homopolymer bases was used. To be conservative, ambiguous bases were counted as errors in Table 4. Based on these data, we chose a cutoff of 50x minimum read depth for high accuracy consensus sequences after error correction with Nanopolish and manual correction of indels in homopolymers.

### 3.3. India

Mission Rabies in partnership with the Government of Goa and Edinburg University identified 104 rabies samples of interest that were collected as part of ongoing canine rabies surveillance, vaccination, and control efforts in Goa, India, during 2016–2018 (Appendix A). Samples included 98 canine, three bovine, two feline, and one jackal identified as positive rabies cases by DFA. RNA was extracted from the 104 samples using the Direct-zol RNA miniprep kit (Zymo Research, Irvine, USA) at the Disease Investigation Unit (DIU), Directorate of Animal Health and Veterinary Services in Goa. Twenty samples were tested by LN34 assay for the presence of rabies virus RNA; 19 samples (95%) were positive. These 19 samples and 85 additional samples that were positive by DFA testing at DIU were selected for nucleoprotein (N) and glycoprotein (G) gene sequencing using the nanopore sequencing protocol. A total of 80 complete N gene and 97 complete G gene sequences were generated (Table 5).

All sequences clustered in the Arctic-like 1a rabies virus lineage (Figure 2), in agreement with previous sequences from Goa, India [84,85]. No sequences fell into the Indian Subcontinent lineage, although a previous non-peer reviewed study identified one isolate from this lineage in Goa in 2014 (Figure 2) [85]. All newly generated sequences from Goa were >99% identical to each other, except for one sequence that was only 96% identical to the other Goa sequences. Further investigation revealed this sample was collected from a dog that was transported from Rajasthan in Northern India into Goa. The remaining sequences were most similar to sequences from canine samples collected in Goa during 2014 [85] and a human case from Goa in 2005 [84] (Figure 2). Clustering in the phylogenetic analysis did not correlate with host animal species (canine, feline, or bovine); rather, sequences from all host species clustered together, suggesting one predominant enzootic transmission cycle in dogs.

### 3.4. Kenya

Twenty-eight canine and bovine samples were collected from Kenyan districts with pilot rabies surveillance and canine vaccination programs in regions near Kisumu, Siaya, Makueni, Kericho, and Nakuru (Appendix A). RNA extraction was performed using the Direct-zol RNA miniprep kit, and samples were tested by the LN34 real-time RT-PCR test prior to sequencing. Nineteen (67.9%) of the samples were positive for rabies virus RNA, eight samples were negative, and one sample was inconclusive by the LN34 test (Table 5, Appendix A). Sequencing of the rabies virus N and G genes was attempted for the 19 positive samples. G gene sequences were produced for 16 samples (84% of positive samples: 11 complete G gene sequences and 5 partial G gene sequences, Table 5). Fourteen N gene sequences were produced (74% of positive samples), including 10 full length N gene sequences and 4 partial sequences (Table 5). Three positive samples with lower rabies RNA levels (LN34 Ct values 26, 28, and 35) did not produce any sequence. Samples that produced full sequences had LN34 Ct values ranging from 15 to 21. Partial sequences were produced from samples with LN34 Ct from 21 to 31.

Preliminary analysis of N and G gene sequences revealed the presence of two groups (Figure 3 red and blue). Comparison with publicly available rabies virus sequences revealed that the new Kenya sequences fell into the Cosmopolitan rabies virus clade, in either the Africa 1a or Africa 1b lineage (Figure 3) [86]. Newly generated full-length Kenya N gene sequences were 99.98% (Africa 1a isolates) and 99.32% (Africa 1b) identical within each group, but only 93.39% identical between groups. Full-length G gene sequences were 99.81% (Africa 1a) and 98.17% (Africa 1b) identical within each group and 93.88% identical between groups. The G gene sequences from Kenya that clustered in the Africa 1a lineage were most similar to reference sequences from Somalia, Ethiopia, and a Polish human case from 1985 (96.92% nucleotide identity on average) (Figure 3). The N gene sequences from samples that clustered with Africa 1a lineage were most similar to reference sequences from Uganda, with >99% identity with a partial N gene sequence from Ntoroko, Uganda, collected in 2010 (GenBank Accession KJ133660.1). The remaining Kenya sequences fell into the Africa 1b lineage and were closely related to previously published sequences from Kenya and Tanzania (with 98.33% nucleotide identity, on average) (Figure 3).

### 3.5. Guatemala

Twenty-five brain samples were collected by the Guatemala National Health Laboratory (LNS) and Ministry of Agriculture, Livestock, and Food (MAGA) during routine rabies surveillance in 2018. Four samples were canine and 21 were bovine. RNA was extracted using the Direct-zol kit (Zymo) at the Guatemala National Health Laboratory (LNS) then transported to the Universidad del Valle de Guatemala in Guatemala City for sequencing. Samples were tested by the LN34 real-time RT-PCR test, and thirteen (52%) were positive for rabies virus RNA, 11 samples were negative, and one sample was inconclusive (Appendix A). Sequencing of rabies virus N and G genes was attempted for the 13 positive samples. Sequencing of the Guatemalan samples posed challenges, and both the number of reads produced and the number of mapped reads were very low for all samples. For this reason, the minimum read depth threshold for this dataset was lowered to 25x to avoid data loss, noting that the consensus sequence accuracy may have been affected. Very few rabies sequences from Guatemala are publicly available, so maximizing data was given priority, especially since we expect accuracy of approximately 99.8% for polished consensus sequences with >25x read depth based on our validation study (Table 4). N gene sequences were produced for six samples (46% of positive samples, Table 5). Four G gene sequences were produced (23% of positive samples, Table 5). Seven positive samples did not produce any sequence (Ct values ranged from 14.4 to 35). Samples that produced sequences had Ct values ranging from 17.1 to 27.4.

Five N gene and four G gene sequences were produced from positive bovine samples. The bovine N gene sequences were 99.5% identical to each other and clustered with rabies virus isolates circulating in the common vampire bat (*Desmodus rotundus*) in Mexico and Guatemala (Figure 4). These new sequences were most similar to isolates from Guatemala 2012 (KF656697), Chiapas, Mexico 2000 (AY854592, antigenic variant AgV11), and Campeche, Mexico (GU991823, antigenic variant AgV11).

An N gene sequence was produced for one canine sample. The N gene from the canine sample shared only 84.6% nucleotide identity with sequences from the bovine samples. The canine sequence grouped within the Cosmopolitan clade of canine rabies virus variants in the Americas-2a lineage, with sequences from dogs from Chiapas, Mexico (Figure 4). The N gene was 99.2% identical to a sequence from a dog collected from Chiapas, Mexico, in 2002 (FJ228518, antigenic variant AgV1).

### 3.6. Vietnam

Twenty-one samples collected as part of routine rabies surveillance programs in Northern and Central Vietnam were selected for sequencing (Appendix A). Twenty samples were from dogs, as part of ongoing active surveillance in Phú Thọ province. One sample was collected from a cow in Quảng Nam province. All samples were previously identified as rabies virus positive by PCR and/or FAT testing at the National Center for Veterinary Diagnosis (NCVD) in Hanoi, Vietnam. All 21 samples (100%) were confirmed positive by the LN34 assay. Rabies virus N and G gene sequences were produced for all 21 samples (100%) (Table 5).

Preliminary sequence analysis of the N and G genes revealed high similarity among the 20 canine samples, with 99.82% and 99.79% average sequence identity, respectively. These 20 sequences clustered in the South East Asia 1 (SEA1) clade (Figure 5). Current publicly available reference sequences in the SEA1 clade include many rabies virus isolates from China and few isolates from Vietnam. The 20 canine-derived sequences generated as part of this study were most similar to rabies virus isolates from Yunnan, Sichuan, and Chongqing, China.

The sequence generated from the positive cow sample was only, on average, 88.66% (N gene) and 85.98% (G gene) identical to the 20 canine-derived sequences. This sequence clustered in the SEA3 canine rabies virus clade and grouped with sequences from Laos (Figure 5). The complete N gene sequence from this bovine sample was only 97% identical to the nearest reference (KX148257 dog from Laos 2002), indicative of divergence.

## 4. Discussion

Many canine rabies endemic countries have begun planning efforts to eliminate dog-mediated human rabies deaths as part of the international “Zero by 2030” campaign [10]; however, effective control and elimination strategies require accurate surveillance information [21]. Laboratory confirmation of rabies cases and rabies virus characterization can provide information about rabies prevalence and distribution. Proper characterization of the diversity and geographic distribution of rabies virus variants in a country can be used to help plan, monitor, and confirm canine rabies elimination, as well as identify cases of rabies importation and track outbreaks. Identification of rabies virus variants, however, is not a standard practice in many countries. In this study, we performed rabies virus gene sequencing in four laboratories in canine rabies endemic countries with varying levels of experience and capacity for molecular methodologies. The data presented here not only provide an example of the successful use of the MinION for portable rabies sequencing but also produced significant epidemiological impact for the regional laboratories visited.

We developed a low-cost portable sequencing method that targets a portion of the rabies virus genome using PCR amplification. Combined nucleoprotein (N) and glycoprotein (G) sequencing increases the resolution and reference comparisons that can be made, compared to partial gene sequencing or sequencing either gene alone. Sequencing of both genes has been performed in several studies [87,88,89]; and although the resolution is lower than whole genome sequencing, N and G gene sequencing has several advantages over whole genome sequencing. First, our method allowed for the use of degenerate primers that work broadly against several diverse lineages of rabies virus, meaning only one set of primers can be used for many samples. Because of the diversity across *Rabies lyssavirus*, whole genome tiled primers are limited to a given lineage, clade, or variant [90,91,92,93], meaning divergent, new, or imported rabies virus variants may be missed. Second, the partial genome method presented here costs much less per sample than a whole genome PCR approach with the MinION [93]. Lastly, the N and G gene sequencing approach is easier to transfer and implement in new laboratories because there are only two amplicons per sample and a new primer design is not required. Whole genome sequencing of rabies virus can alternatively be performed by directly sequencing viral cDNA or RNA; however, this approach is much more costly because it is difficult to multiplex. The use of PCR in our approach amplifies rabies signal above the host signal and allows for multiplexing up to 96 samples in a single sequencing run, which lowered the cost of sequencing to USD 15 when 50 samples were sequenced at once.

In Goa, India, we tested this approach for high throughput. We were able to sequence over 100 samples in five days with just two operators and all portable, low-throughput equipment; however, RNA extraction had been performed previously. In Kenya, fewer samples were available, but we were able to extract RNA, perform rabies diagnostic testing by PCR, sequence positive samples, and provide genotyping and phylogenetic analysis during a short five-day trip with a single operator performing wet lab and bioinformatic analysis. In Guatemala, we observed the highest failure rate. The low success was most likely caused by low sample quality (either tissue or RNA), although PCR machine error was also observed. Additionally, the primers used in this study performed better for canine rabies virus lineages than bat lineages, which are highly diverged. In Kenya, samples that did not produce sequences exhibited lower levels of viral RNA, as determined by real-time PCR, or exhibited firmness and moisture similar to formalin-fixed tissue. In Guatemala, RNA levels did not correlate with sequencing success. If RNA in those samples was fragmented due to degradation, real-time PCR, which amplifies a very short portion of the genome, would be more successful than the long amplicon PCR that was used for sequencing. In cases of sample degradation and suspected RNA fragmentation, sequencing partial gene amplicons may be preferable to complete gene sequencing, and this protocol can be easily modified for use with any rabies primers. Storage of tissue or RNA in stabilization buffer or at −80 °C is recommended until samples can be processed for sequencing. Future work is needed to expand this method for broad use across rabies virus variants and in samples of poor quality or low viral RNA load.

It is thought that more human rabies deaths occur in India than in any other country in the world, with more than 20,000 estimated deaths per year [3,94]. Previous studies have identified the circulation of Arctic-like, Cosmopolitan, and Indian Subcontinent rabies virus lineages throughout India [94,95,96,97,98,99]. Despite the high disease burden, rabies is neglected in India, and there is currently no nation-wide, government-funded canine vaccination effort. Recently, a few cities and states in India have initiated mass dog vaccination strategies to control canine rabies [100,101,102,103,104,105]. Goa has been a test case for canine rabies elimination in India, with striking drop-offs in human and canine rabies cases after multiple canine vaccination campaigns and extensive surveillance in recent years [104]. As Goa moves toward elimination of canine rabies, Mission Rabies and their collaborators became interested in rabies virus sequencing, but there was no local sequencing capacity, and political restrictions prevented shipment of samples out of the country. Portable sequencing with the MinION allowed for in-country sequencing of these high-priority samples in a veterinary diagnostic laboratory that lacked molecular biology laboratory equipment. All sequences generated from Goa belonged to the Arctic-like 1a canine rabies virus lineage (Figure 2), in agreement with sequences from canine isolates from Goa in 2014 and an isolate from a UK traveler to Goa in 2005 [84,85]. None of the almost 100 samples sequenced in the current study were of the Indian Subcontinent lineage, and the genetic diversity of isolates collected was much lower than for samples collected in 2014.

In Kenya, we identified circulation of Africa 1a rabies virus lineage. Previously, the limited available rabies virus sequences from Kenya had all belonged to the Africa 1b lineage [31,86,106]. This finding is being combined with epidemiological data to try to understand the distribution of these two rabies lineages in Kenya, which will be useful in the ongoing rabies vaccination campaigns and elimination strategies [93]. Based on the data generated in this study, the two lineages may be separated geographically by Lake Victoria (Figure 3); however, the limited sample size prevents any definitive conclusions. Future studies with increased sample size are needed to determine if the Africa 1a and 1b lineages overlap in their distribution.

Among of the major and most important reservoirs of rabies virus in Latin America is the common vampire bat, *Desmodus rodundus* [107]. Rabies in vampire bats has been reported throughout Latin America, from Mexico to Argentina [108]. Vampire bat rabies represents a major economic impact to cattle, while also imposing health risks to humans and other domestic animals [109]. Both domestic dogs and vampire bats act as rabies reservoirs in Guatemala, with dogs accounting for >70% of human cases from 1994 to 2012 [89]. However, the characterization of rabies virus variant responsible for human and animal cases is not routinely performed in Guatemala, so the prevalence of canine or vampire bat rabies are not well understood. Only a single study has published rabies virus sequences from Guatemala; and those sequences are from two isolates collected from vampire bats [89]. In this study, we produced sequences for five additional Guatemalan vampire bat rabies variant isolates, this time isolated from cattle, and we present the first publicly available sequence of Guatemalan canine variant isolated from a rabid dog.

Rabies does not respect borders [2], and several previous studies have identified and traced rabies re-emergence in rabies-free regions and spread of rabies across political borders [19,110]. Importation of rabies into rabies-free areas can be extremely expensive [110,111,112,113,114,115,116,117]. In Goa, we identified one suspect translocation of a rabid dog from Northern India into Goa. Previously, translocation of a rabid dog from Northern India (Punjab) to the eastern coast of Southern India (Hyderabad, Andhra Pradesh) was also identified by sequencing; rabies translocation may be common in India due to unrestricted movement of animals [97]. Goa is rather isolated as it is surrounded by the Arabian Ocean on the west and the Western Ghats on the east; however, this case shows that even for isolated regions, the risk of canine rabies re-introduction is very serious.

In our study, we found that all 20 canine rabies samples from Phú Tho, Vietnam, grouped in the SEA1 rabies virus clade and were most similar to rabies virus sequences from the Yunnan, Sichuan, and Chongqing regions of China. The majority of previously published sequences from Vietnam belong to the SEA3 clade [31,32,45,88,118,119,120], although a few published sequences from Northern Vietnam group in the SEA1, SEA5, and Cosmopolitan canine rabies virus clades [31]. Both Vietnam and China reported rabies epidemics between 2007 and 2010 [118,120,121,122]. In China, the SEA1 rabies clade was the dominant clade to emerge during the epidemic [122] and displace the SEA2 and SEA3 clades in Yunnan Province, China, which shares a border with Vietnam [45]. The high genetic identity between the Vietnam isolates sequenced in this study and those from Southern China suggest potential spread of canine rabies across the China–Vietnam border. Two previous studies have found evidence of rabies spread across the China–Vietnam border [118,119], which has been attributed to cross-border dog movements [45,120]. In contrast, the rabies virus isolated from a cow in Quảng Nam was most similar to rabies virus sequences from Vietnam and Laos; however, this sequence was only 97% identical to the closest available reference sequences from Laos, indicating divergence and suggesting it was not likely directly related to rabies virus circulating in Laos based on the publicly available data. In such cases where sequence evidence is suggestive of potential cross-border spread, future monitoring by both countries is important as are coordinated canine vaccination campaigns and/or restriction on animal movements.

Lastly, it is well accepted that MinION sequencing can produce errors in homopolymer regions that cannot currently be resolved bioinformatically. In this study, we used manual correction of homopolymer regions to attain the highest quality consensus sequences. The rabies virus genome contains only five genes, and each one is essential for viral function. Single nucleotide indels in homopolymer regions produced by MinION sequencing cannot be corrected by high read depth or error correction. In the coding region of a gene, this type of mutation would cause a frame shift. The likelihood of a frame shift mutation naturally occurring in the nucleoprotein or glycoprotein genes in a viable virus is exceedingly low. Given these observations, we reason that manual correction of indels in homopoymers is appropriate to resolve homopolymer indel errors within the coding region of the nucleoprotein and glycoprotein genes in rabies virus. Manual error correction is likely not appropriate for non-coding regions or coding regions of most organisms and must be evaluated by subject matter experts for use in other systems.

## Figures and Tables

**Figure 1 viruses-12-01255-f001:**
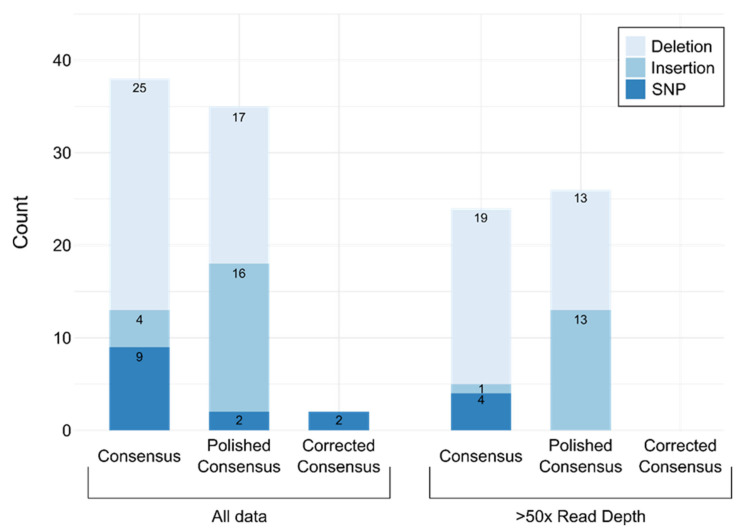
Errors observed in consensus sequences generated from MinION sequencing data by three methods. Number of differences in the nucleoprotein gene (1353 nt) for raw consensus (Raw Consensus), consensus after polishing with raw reads using Nanopolish (Polished Consensus), and after manual correction of indels in homopolymer regions (Manual Correction) compared to consensus generated by Sanger sequencing is shown. Insertions, deletions, and single nucleotide changes (miscalled bases or SNPs) are shown in different shades of blue. Number of each type of error is depicted on the bars. Data shown include data from all sequences (left) and those sequences produced with >50x read depth (right).

**Figure 2 viruses-12-01255-f002:**
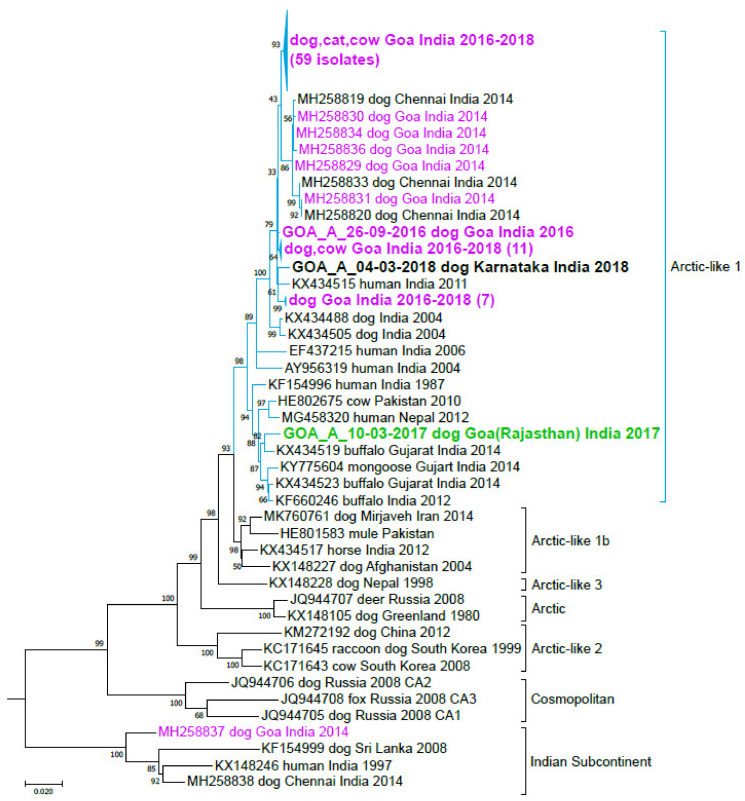
Phylogenetic analysis of MinION sequences generated in Goa, India. Complete nucleoprotein gene sequences from Goa rabies virus isolates were compared to publicly available sequences. Reference sequences were chosen based on sequence similarity to the newly generated sequences or inclusion in rabies Arctic-like 1, Arctic, Arctic-like 2, Arctic-like 3, Indian Subcontinent, and Cosmopolitan phylogenetic clades (from [31]). Newly generated sequences are shown in bold; several sequences are collapsed for viewing ease. Sequences from Goa isolates are shown in magenta. Translocation case from Rajasthan is highlighted in green. Accession number, host, location, and collection date are included for reference, when available; additional sample details can be found in Appendix A. Phylogenetic analysis was performed by maximum likelihood based on a GTR+G+I model. Differences between samples are shown by the number of changes per site along the horizontal axis. Bootstrap values near the branch points represent the percentage of trees that had the same clustering out of 1000 replicates.

**Figure 3 viruses-12-01255-f003:**
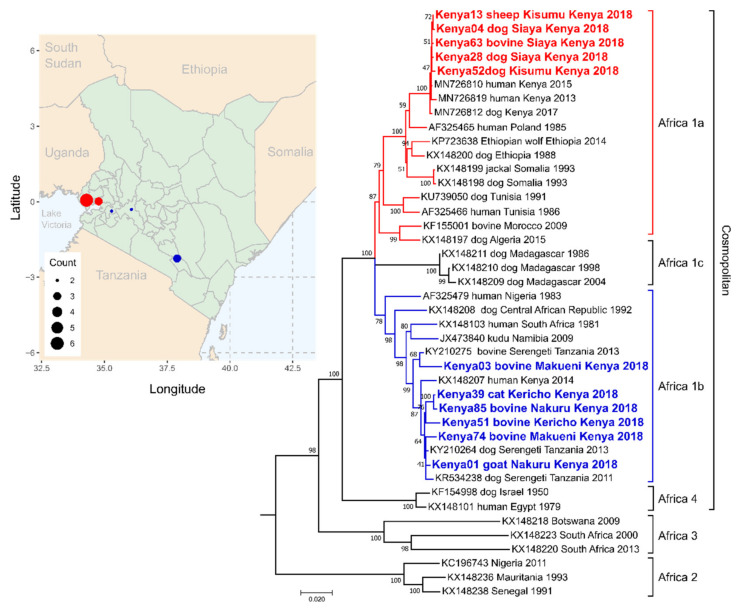
Phylogenetic analysis of MinION sequences generated in Nairobi, Kenya. Full-length glycoprotein gene sequences (1575 nt) from Kenya rabies virus isolates were compared to publicly available sequences. Reference sequences were chosen based on sequence similarity to the new Kenya sequences or inclusion in Africa 1a, Africa 1b, Africa 1c, Africa-2, Africa-3, or Africa 4 rabies phylogenetic clades based on Troupin et al. [31]. Accession number, location, and collection date are included, when available. Newly generated sequences are highlighted in color, corresponding to rough location on the map to the left. Additional sample information can be found in Appendix A. Phylogenetic analysis was performed by maximum likelihood based on the GTR+G+I model. Number of changes per site is shown along the horizontal axis. Bootstrap values near the branch points represent the percentage of trees that had the same clustering out of 1000 replicates.

**Figure 4 viruses-12-01255-f004:**
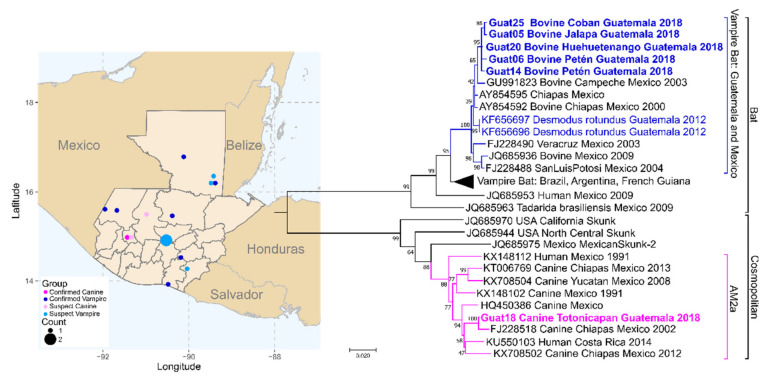
Phylogenetic analysis of MinION sequences generated in Guatemala City, Guatemala. Complete nucleoprotein gene sequences from Guatemala rabies virus isolates were compared to publicly available sequences. Reference sequences were chosen based on sequence similarity to the new Guatemala sequences, isolation from vampire bat or bovine in Mexico or Guatemala, or inclusion Cosmopolitan Americas-2a (AM2a) rabies phylogenetic clade based on Troupin et al. [31]. Accession number, host animal, location, and collection year are included for reference, when available. Sequences generated in this study are highlighted in bold; additional information can be found in Appendix A. Colored points on the map correspond to the location of LN34 positive samples; confirmed canine or vampire bat lineage isolates are colored dark blue or dark pink on the map and phylogenetic tree. Suspected lineage was based on host (canine or bovine). Phylogenetic analysis was performed by maximum likelihood based on the GTR+G+I model in Mega7. Number of changes per site is shown along the horizontal axis. Bootstrap values near the branch points represent the percentage of trees that had the same clustering out of 1000 replicates.

**Figure 5 viruses-12-01255-f005:**
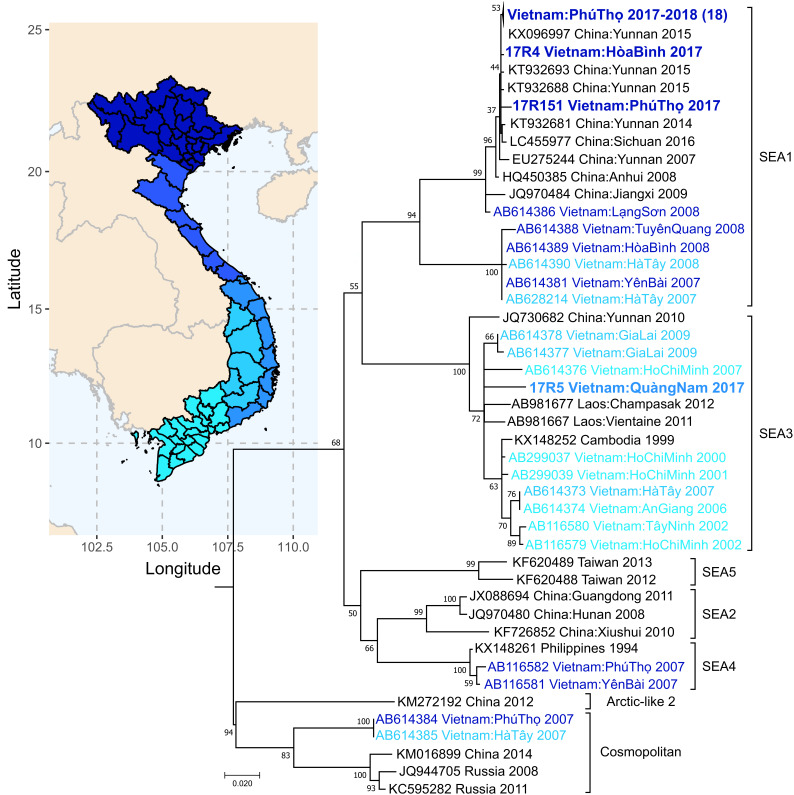
Phylogenetic analysis of MinION sequences generated in Hanoi, Vietnam. Complete nucleoprotein gene sequences from Vietnam rabies virus isolates were compared to publicly available sequences. Reference sequences were chosen based on sequence similarity to the new Vietnam sequences or inclusion in South East Asia 1–5 (SEA1–SEA5) or Cosmopolitan rabies phylogenetic clades based on Troupin et al. [31]. Sequences generated in this study are highlighted in bold; some sequences have been collapsed for ease of viewing. Color coding of sequence names correspond to region location in Vietnam as colored on the map. Accession number, host animal, location, and collection date are included for reference, when available; additional information can be found in Appendix A. Differences between samples are shown by the number of changes per site along the horizontal axis. Bootstrap values near the branch points represent the percentage of trees that had the same clustering out of 1000 replicates.

**Table 1 viruses-12-01255-t001:** Primers used in this study. Sequences are in 5′ to 3′ orientation. Red colored bases correspond to the rabies virus genome binding sequence. Black nucleotides at the 5′ end were used to add barcode sequences for multiplexing. Red sequences correspond to positions 54 to 76 (NgeneFor), 1475 to 1496 (NgeneRev), 3273 to 3300 (GgeneFor), and 5264 to 5288 (GgeneRev) in rabies virus strain PV-2061 (accession number JX276550).

Primer	Sequence
NgeneFor	TTTCTGTTGGTGCTGATATTGCAATGTAACACCYCTACAATGGAT
NgeneRev	ACTTGCCTGTCGCTCTATCTTCAGGAGGRGTGTTAGTTTTTTTC
GgeneFor	TTTCTGTTGGTGCTGATATTGCGATGTGAAAAAAACTATYAACATCCCTC
GgeneRev	ACTTGCCTGTCGCTCTATCTTCTGTGAKCTATTGCTTRTGTYCTTCA

**Table 2 viruses-12-01255-t002:** Portable equipment used in this study.

MinION (Oxford Nanopore Technologies, Oxford, UK)
Mic qPCR cycler (Bio Molecular Systems, El Cajon, CA, USA)
MiniPCR (Amplyus, Cambridge, MA, USA)
E-Gel Electrophoresis System (ThermoFisher, Waltham, MA, USA)
Mini Centrifuge (Southern Labware, Cumming, GA, USA)

**Table 3 viruses-12-01255-t003:** Price of consumable reagents for MinION sequencing of rabies virus amplicons. Prices are based on reagents purchased in the United States as advertised on the Oxford Nanopore website at the time of sequencing (2018). Bulk price of USD 500 per flow cell was used. Shipping prices are not included. Prices may vary substantially depending on location and product availability.

	Price/Sample		
	50 Samples/Run	20 Samples/Run	1 Sample/Run
PCR Barcoding	USD 2.11	USD 2.11	USD 0.00
Library Preparation	USD 2.79	USD 6.57	USD 126.13
Sequencing	USD 10.00	USD 25.00	USD 500.00
Total	USD 14.90	USD 33.68	USD 626.13

**Table 4 viruses-12-01255-t004:** Accuracy of consensus sequences generated from MinION sequencing data by three methods. Sequence accuracy of the raw consensus (Raw), consensus after polishing with raw reads using Nanopolish (Polished), and Nanopolished consensus after manual correction of indels in homopolymer regions (Manual) is shown relative to Sanger sequencing consensus. Sequence accuracy and number of sequences examined are shown for six different read depth levels. Accuracy above 99.99% is highlighted in red text.

Read Depth	>0	<50	>50	>100	>1000	>10,000
Sequences	70	11	58	48	13	5
Raw	99.842	99.213	99.958	99.968	99.965	99.958
Polished	99.933	99.778	99.961	99.965	99.987	100
Manual	99.988	99.940	99.996	99.999	100	100

**Table 5 viruses-12-01255-t005:** Success rate of MinION sequencing in Guatemala, India, Kenya, and Vietnam. The total number of positive samples was based on the result of diagnostic testing in the local laboratory; for Guatemala, Kenya, and Vietnam, all positive samples were confirmed by the LN34 real-time PCR assay. Only 19 samples from India were confirmed positive by LN34 PCR (see Appendix A for additional details). The number and percent of positive samples where N or G gene sequences were produced is shown for each country.

Country	Samples	Positive	N full	G full	N Partial	G Partial	Percent Success
Guatemala	25	13	6	4	0	0	46.15
India	104	103	80	97	0	0	93.27
Kenya	28	19	10	11	4	5	84.21
Vietnam	21	21	21	21	0	0	100.00

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
