# Peer review of "Portable Rabies Virus Sequencing in Canine Rabies Endemic Countries Using the Oxford Nanopore MinION"

_viruses, 2020, doi:10.3390/v12111255_

Round 1

Reviewer 1 Report

The manuscript of Gigante et al describes implementation of MinION sequencing of rabies virus genes from field samples in the limited resource settings. In fact, this is a development of the recently published study (https://pubmed.ncbi.nlm.nih.gov/32090172) with partial participation of the same co-authors, and more generally lies in line of papers dedicated to the use of MinION for sequencing of pathogen genes. The advantages of the method are the relatively low cost and portability, but disadvantages include low sensitivity and high error rate.

The major issue that I see in this specific study is the low sensitivity that authors encountered trying to amplify the entire N and G genes of rabies virus. It is true that the entire gene sequences provide a better resolution for phylogenetic (and other genome-based) studies and add a greater value to the available collection of rabies virus genes. However, the high error rate which often require manual correction (which from my prospective is also not error-free, particularly if performed by not a very experienced analyst) diminish the utility of the generated data. On the other hand, low sensitivity of the approach resulted in inability to gain any sequence information from a substantial proportion of samples (Table 5). Although in “Discussion” authors suggest that this problem may be related to the variability of rabies virus genomes resulting in primer mismatches, my experience suggests that primers used by authors will amplify all rabies virus variants present in the study. As another plausible explanation I suggest the degradation of viral RNA in the field samples collected in hot environment that is common in the limited-resource countries (including the countries covered by the present study: India, Kenya, Guatemala). In such settings PCR-amplification of shorter partial N gene sequences may be successful and provide sufficient input for surveillance and control needs. Although this would not be as comprehensive as the entire N (and other genes), it is better than no genetic information at all, and from this prospective may outperform the method implemented by authors. Indeed, authors amplified not the entire viral gene in one RT-PCR but rather relied on 2-3 overlapping fragments. However, if RNA is degraded, the length of the amplified fragment may be shortened further (e.g. the region encompassing first 400 nucleotides of N gene was successfully used by many European authors during decades and satisfied the global phylogenetic needs). Although this is a testable hypothesis, I do not suggest that it could be evaluated by authors of this manuscript, given the amount of work it requires with all international activity involved. But as the minimum I strongly recommend that such possibility is elaborated in “Discussion” and recommended for further evaluation.

Therefore, I have only minor comments left:

Line 53: a disease cannot be cause by virus species, only by a virus.

Line 54: Lyssavirus must be italicized.

Line 55: Rabies lyssavirus must be italicized. The end of the sentence “commonly referred to as rabies virus” is incorrect. Rabies virus is a virus whereas Rabies lyssavirus  is a species.

Line 87: insert “gene” or “genome” between “rabies virus” and “sequencing”.

Line 120: add “genes” after “glycoprotein”.

Line 121: add “genes” after “rabies virus”.

Lines 129, 132” replace “normal” with something more reasonable, e.g. “regular”, “routine”.

Lines 218-219: am curious why the “manual inspection of alignment” (I assume visual inspection rather than manual) was implemented and which sequences were corrected (MinION or Sanger) as a sophisticated algorithm for error correction was described above in the sub-section 2.4?

Sub-section 2.11: cost estimate: it appears that some items essential for the analysis performed (and non-essential for simple RT-PCR testing and for Sanger sequencing) are missing, e.g. Geneious software (in the USD for the US territory) costs $1,500 per seat/year, Qubit costs ~$4,000. As well as reagents. All these items add substantial cost in resource limited settings.

Line 280: The 15% error rate is really high. I cannot understand realistically how such rate may occur if coverage is sufficiently high (e.g. >1000 per site), which is expected in PCR-amplified samples, and the errors occur randomly. It seems that more than every other basecall must be wrong.

Line 288: insert “positive” after “rabies”.

Lines 288-296; 340-343, 439-442: Again, am surprised with such low coverage given that authors sequenced PCR-amplified genetic material. Will appreciate an explanation for this. This seems to be an unusual situation as other authors reported successful MinION sequencing of ssRNA virus genes from field samples without PCR-amplification (e.g. https://pubmed.ncbi.nlm.nih.gov/32523561, https://pubmed.ncbi.nlm.nih.gov/31322493, https://pubmed.ncbi.nlm.nih.gov/30858830, several papers on influenza A viruses).

Line 302: the manual correction of the consensus can be achieved only via template-based assembly. As the gold standard of sequencing is de-novo assembly of contigs, the approach per se is suboptimal however, for the purpose of this study is justified (lines 303-309). Nevertheless, I’m again quite surprised that PCR-amplified products which should provide an abundant coverage of target genome regions even after all “polishing” manipulations produced indels in the consensus contigs.

Line 324: “the only remaining errors were miscalled bases” is actually a significant issue. Particularly if they are non-synonymous, and if after GenBank deposition will be used by further authors as reference sequences. Even from phylogenetic prospective, a single fixed substitution may distinguish may distinguish one viral lineage from another. More importantly, when binding sites for molecular drugs monoclonal antibodies are evaluated, their amino acid composition and structure may be determined incorrectly if based on erroneous sequences. Authors of this study may be aware of this and provide relevant comments in GenBank submissions but authors of further similar studies may not do the same which will accumulate the number of erroneous sequences in GenBank.

Line 369: Am quite surprised that LN34 qPCR assay confirmed only 19 samples from India. Am I correct that all 104 samples were tested by the LN34 and only 19 of these were positive? If so, it appears that the assay is inadequate for rabies viruses circulating in India. Or, only 19 samples were tested by the LN34? Authors need to clarify and elaborate this observation. Table 5 shows that 80 N genes and 97 G genes were sequenced which demonstrates that these samples in fact were positive.

Figure 2: it appears to me that sequences GU937036 and GU937041 originate from North Korea. In addition to representatives of the cosmopolitan rabies virus lineage, I would like to see in this tree some sequences from the Indian Subcontinent lineage (to highlight geographic diversity of the viruses).

Lines 404-406: I do not consider that Ct values 26-28 correspond to low viral RNA loads, particularly as authors used RT-PCR amplification prior to MinION sequencing (and authors state in line 406 that from samples with Ct=31 they could obtain partial sequences, and this is at least 10x lesser RNA load that at Ct=27-28). Perhaps the integrity of RNA in these samples was insufficient to produce long reads that were targeted by the RT-PCR primers. Alternatively, sustainability of MinION sequencing appears quite questionable. Poor integrity of ssRNA viral genomes in field samples, particularly those obtained in hot environment (frequently met in resource-limited countries of Southern Asia and Africa) suggest that short partial sequences of the most abundant rabies virus N gene should be targeted not only by RT-PCR amplification but also by MinION sequencing for phylogenetic purposes. Better to have a shorter sample than no sample.

Figure 4: As the tree includes viral lineages Africa-1, 3, and 4, I see logical to include a few representatives of lineage Africa-2.

Line 434: replace “normal” with “regular” or “routine”.

Lines 439-449: as in the comments to lines 404-406 above, it seems that RT-PCR amplification of long sequences might be compromised by RNA deterioration, and as such, shorter portions of N gene would constitute a compromise to fulfil the phylogenetic surveillance needs.

Lines 520-526: This is true that sequencing of two genes, N and G, provides a better resolution than sequencing of shorter genome regions. However, given the limited sensitivity of MinION sequencing even after preliminary RT-PCR amplification of the samples, and frequent inability to generate sequences from samples with adequate viral RNA load (likely due to RNA deterioration), I strongly recommend to revise the approach, and consider RT-PCR amplification of partial N genes with their further MinION sequences. Numerous previous studies demonstrated that such approach will still be sufficient for the majority of phylogenetic applications (and result in the same topology of phylogenetic trees as depicted in the Figures 2-5).

Lines 524-525: given error-prone MinION sequencing, I would avoid such broad scope of sequence use (e.g. unlikely suitable for evolutionary analysis and host adaptation) but rather focus on phylogenetic needs which may facilitate surveillance and control programs.

Lines 541-543: As authors indicated in line 208, they used primers Lys 001-1066degB and 550F-304 which, indeed, were constructed successfully used previously for many diverse rabies virus variants, including all “terrestrial” and chiropteran American viruses. Therefore, poor performance of the method for Guatemalan samples cannot be attributed to primer mismatches. I think authors correctly refer this issue to poor sample quality (lines 545-550) as I also mentioned above in the comments to lines 404-406, 520-526, and in the major comments on top of my review.

Lines 552-553: am curious which primer optimization was required (and performed) “after the trip”?

Line 594: replace “the principle” with “one of the major and most important”.

Lines 506-672: “Discussion” is inappropriately long, may be shortened at least ~30% with better focus on the findings of this specific study and elimination of excessive common musings. I also suggest to include in “Discussion” my major comment on amplification and sequencing of shorter fragments of N gene.

Reviewer 2 Report

This paper describes the use of oxford Nanopore MinION to sequence rabies virus samples from a number of developing countries. It has demonstrated the quality and utility of the technology in the regions of the world that do not have well-established diagnostic infrastructure.  The technology is promoted as a low-cost solution; however, this point is not fully established. The calculation of the costs only included reagents and was based on the price in US. A large number of samples have to be sequenced for it to be cost effective. Many other costs and factors that will affect the application of the technology have not been discussed. Nevertheless, the paper has provided some useful information.

Minor text editing is needed.

Author Response

We thank the reviewer for his/her comments. We have updated the language to make it clearer that the cost estimate presented is for consumable reagents only and not the cost of implementing sequencing, which can be a prohibitive depending on the resources of the laboratory. A list of the equipment and acknowledgement of the expertise required has been added to avoid misleading readers. The low-cost claim was meant to be compared to other next generation sequencing approaches and to whole genome sequencing.